# IFN-γ Critically Enables the Intratumoural Infiltration of CXCR3^+^ CD8^+^ T Cells to Drive Squamous Cell Carcinoma Regression

**DOI:** 10.3390/cancers13092131

**Published:** 2021-04-28

**Authors:** Zhen Zeng, Margaret Veitch, Gabrielle A. Kelly, Zewen K. Tuong, Jazmina G. Cruz, Ian H. Frazer, James W. Wells

**Affiliations:** 1The University of Queensland Diamantina Institute, Faculty of Medicine, The University of Queensland, Brisbane, QLD 4102, Australia; jenny.zeng@uq.edu.au (Z.Z.); m.veitch@uq.edu.au (M.V.); gabbyk@cleanairaust.com.au (G.A.K.); j.gonzalezcruz@uq.edu.au (J.G.C.); i.frazer@uq.edu.au (I.H.F.); 2Department of Medicine, University of Cambridge, Cambridge CB2 0QH, UK; zkt22@cam.ac.uk; 3Dermatology Research Centre, The University of Queensland, The University of Queensland Diamantina Institute, Brisbane, QLD 4102, Australia

**Keywords:** squamous cell carcinoma, IFN-γ, CD8 T cell, regression, immune control

## Abstract

**Simple Summary:**

Cutaneous squamous cell carcinoma (SCC) is prevalent in aged individuals and individuals with compromised or weakened immune systems, indicating a close association between immune function and SCC control. The aim of our study was to uncover the identity of key immune subsets that mediate SCC control, and to elucidate the mechanistic role of the proinflammatory cytokine Interferon-gamma in this process. We established a SCC regressor model, which we used to determine that: (1) CD8^+^ T cells, not CD4^+^ T cells or NK cells, are essential for SCC regression; (2) the neutralisation of Interferon-gamma prevents CD8^+^ T cell infiltration and SCC regression; (3) CD8^+^ T cell migration into SCC critically depends upon Interferon-gamma-induced chemokine expression. Thus, our model can be used to understand the key immune mechanisms involved in SCC regression, which will support targeted investigations into the integrity of these mechanisms in patients with progressive disease.

**Abstract:**

Ultraviolet (UV) radiation-induced tumours carry a high mutational load, are highly immunogenic, and often fail to grow when transplanted into normal, syngeneic mice. The aim of this study was to investigate factors critical for the immune-mediated rejection of cutaneous squamous cell carcinoma (SCC). In our rejection model, transplanted SCC establish and grow in mice immunosuppressed with tacrolimus. When tacrolimus is withdrawn, established SCC tumours subsequently undergo immune-mediated tumour rejection. Through the depletion of individual immune subsets at the time of tacrolimus withdrawal, we established a critical role for CD8^+^ T cells, but not CD4^+^ T cells, γδ T cells, or NK cells, in driving the regression of SCC. Regression was critically dependent on IFN-γ, although IFN-γ was not directly cytotoxic to SCC cells. IFN-γ-neutralisation abrogated SCC regression, significantly reduced CD8^+^ T cell-infiltration into SCC, and significantly impaired the secretion of CXCL9, CXCL10 and CCL5 within the tumour microenvironment. A strong positive correlation was revealed between CXCL10 expression and CD8^+^ T cell abundance in tumours. Indeed, blockade of the CXCL10 receptor CXCR3 at the time of tacrolimus withdrawal prevented CD8^+^ T cell infiltration and the regression of SCC. Chimeric models revealed an important role for immune cells as producers of IFN-γ, but not as recipients of IFN-γ signals via the IFN-γ receptor. Together, these findings suggest a key role for IFN-γ in driving the expression of chemokines within the tumour environment essential for the destruction of established SCC by CD8^+^ T cells.

## 1. Introduction

Cutaneous squamous cell carcinoma (SCC) is the second most prevalent type of non-melanoma skin cancer, presenting with an estimated >1 million cases per year in the USA alone [1]. These tumours typically arise in aging individuals, or in people who are immunocompromised in some way [2]. Indeed, solid organ transplant recipients suffer a 65–100-fold increased incidence of SCC as a consequence of the immunosuppressive medications they must take to prevent organ rejection [3,4,5], highlighting the strong association between immune function and the emergence of SCC. Several groups including our own have reported that SCC tumours contain a strong abundance of immune cells, inflammatory cytokines, and target antigens [6,7,8,9,10,11], and can therefore be considered to be immunologically ‘hot’ tumours. However, despite the presence of an inflammatory environment, SCC tumours rarely regress, and the standard of care is surgical removal in order to prevent progression to metastatic disease. Each year, the cost of treatment in the USA is estimated to be well over 1 billion dollars [12].

The main determinant for SCC development is cumulative exposure to Ultraviolet (UV) radiation present in sunlight. UV is both mutagenic and immunosuppressive, leading to DNA mutations and the suppression of local immune function. While SCC remains an uncommon tumour subtype in mice, UV-induced tumours in general are highly immunogenic, and are frequently rejected when adoptively transferred into immune competent recipients [13,14,15]. By studying these so called ‘regressor’ tumours, a clear and common role has emerged for CD8^+^ T cells in mediating UV-induced tumour rejection [16]. Infrequently, regressor tumours can develop to become ‘progressor’ tumours that grow in immune competent mice, and the majority do so without loss of target antigen or MHC expression [17,18]. This suggests that loss of tumour control is not due to immunoediting or impaired antigen presentation. However, the mechanisms by which CD8^+^ T cells control regressor tumours, or, presumably, lose control of progressor tumours, are still unclear.

The presence of an inflammatory microenvironment within SCC lesions is well documented [7,19,20]. IFN-regulated gene transcription is increased and correlates with the extent of immune cell infiltration, suggesting that IFN-associated immune responses play an important role in SCC immunosurveillance [19]. Supernatant derived from SCC tumours is known to increase the frequency of CD8^+^ T cells that produce IFN-γ [20], and indeed the quantitation of IFN-γ in the serum of SCC patients is reportedly a useful indicator of tumour progression [21]. Moreover, the incidence of 3-methylcholanthrene-induced skin tumours are accelerated in IFN-γ-deficient mice [22,23]. However, IFN-γ is a pleiotropic cytokine with potential for both protumour and antitumour effects [24], and just how it may contribute to SCC control remains to be elucidated.

The aim of this study was to define the role of IFN-γ in the immune-mediated rejection of SCC. Piecing together the specific mechanisms employed by the immune system to control SCC in mice, should allow for a targeted investigation into the integrity of these mechanisms in patients with progressive disease. In this study, we generated a SCC regressor tumour model in which SCC grown in mice immunosuppressed with tacrolimus are subsequently destroyed by the immune system following the removal of immunosuppression, permitting an examination of the role played by IFN-γ in the rejection of SCC.

## 2. Materials and Methods

### 2.1. Mice

HPV38E6E7-FVB mice, expressing the two viral oncogenes E6 and E7 of Human Papillomavirus (HPV) type 38 under the control of the Keratin 14 promoter [25], HPV38E6E7-FVB × C57BL/6 F1 mice, IFN-γ^−/−^ mice, and IFN-γR^−/−^ mice were bred and kept under standard laboratory conditions at the Translational Research Institute Biological Research Facility (Brisbane, Australia). C57BL/6J mice were purchased from the Animal Resources Facility (Perth, Australia). Mice used in this study were 6–12-week old females/males. All animal procedures were performed with approval from the University of Queensland Animal Ethics Committee (Approval Numbers UQDI/466/14 and UQDI/512/17).

### 2.2. Diet-Based Drug Delivery

Tacrolimus (TAC; LC Laboratories, Woburn, MA, USA) infused mice diet was manufactured by Specialty Feeds (Perth, WA, Australia) as described in our previous study [26]. To ensure minimal degradation, the final product was double-contained inside an airtight bag and stored at 4 °C protected from light. Diet was restocked every 3–4 days.

### 2.3. Establishment of the SCC Mouse Model

A UV-induced SCC cell line derived from HPV38E6E7-FVB mice was created in-house (HPV38 SCC cell line) and cultured in modified Ham’s F12 media (Thermo Fisher Scientific, Waltham, MA, USA) as mentioned in previous research [27]. The HPV38 SCC cell line is a regressor cell line, which is unable to grow in immune-competent animals. HPV38E6E7-FVB mice were thus immunosuppressed by being placed on the TAC-infused diet for 7–8 days prior to subsequent (day 0) with SCC cells on the lower back (subcutaneously s.c. injection, 10^6^ cells/mouse in 100 μL PBS). Mice were maintained on the TAC-infused diet (until removal if relevant as indicated in the figure legends) and tumour growth was recorded every 3–5 days by measuring the major dimension (D) and minor dimension (d) of the tumour via a digital calliper. Measurements were transformed into tumour volume using the formula: tumour volume (cm^3^) = [D × d^2^]/2. Euthanasia was performed when the tumour reached a maximum volume of 1 cm^3^. Histology was performed by the QIMR Berghofer Histology Facility (Brisbane, Australia).

### 2.4. In Vitro IFN-γ Treatment

HPV38 SCC cells were plated on 6-well-plates overnight in modified Ham’s F12 medium at 37 °C, 5% CO_2_. This was followed by treatments with recombinant mouse IFN-γ (10 ng/mL or 100 ng/mL) (R&D Systems, New South Wales, Australia), in fresh complete medium for 72 h. Floating and adherent cells were harvested for Annexin V (Biolegend, San Diego, CA, USA) and 7-AAD (BD Biosciences, San Jose, CA, USA) co-staining to assess the viability of SCC cells after IFN-γ treatment, or for H-2K^q^ (Biolegend) staining to detect MHC class I expression. Supernatants of culture medium were collected to determine CXCL10 and CXCL11 using DuoSet^®^ ELISA kits (R&D Systems) following manufacturer’s instructions.

### 2.5. In Vivo Treatments

In in vivo depletion experiments, 250 μg of the following monoclonal antibodies, purchased from BioXCell (West Lebanon, NH, USA), were injected intraperitoneally (i.p.) in 200 μL PBS: anti-CD4 (clone GK1.5), anti-CD8β (Lyt 3.2; clone 53–5.8), anti-TCRβ (clone H57-597 (also known as HB218)), anti-TCRγδ (clone UC7-13D5), and anti-Gr-1 (clone RB6-8C5). To deplete natural killer cells (NKs), 20 ul of anti-asialo GM1 (Wako Chemicals, VA, USA) antiserum was injected i.p. in 200 μL PBS. The efficacy of depletion was verified by staining of blood cells. Neutralisation of IFN-γ or blocking of CXCR3 signalling was achieved with 200 μg of anti-IFN-γ (clone XMG1.2; BioXCell, Lebanon, NH, USA) or with 300 μg of anti-CXCR3 (CD183; clone CXCR3-173; BioXCell, Lebanon, NH, USA) respectively, in 200 μL PBS (i.p.).

### 2.6. Schedule of In Vivo Depletion Experiments

To examine the influence of immune cell subsets on SCC establishment, specific immune cell subsets were depleted prior to, and immediately following, SCC challenge on day 0. Antibody-depletion injections were initiated on day 4, or in the case of anti-asialo GM1, on day 2. Injection of anti-CD4, anti-CD8β, anti-TCRβ or the appropriate isotype antibodies were repeated on day 3; anti-TCRγδ was repeated on days 3 and 12; and anti-Gr-1 and anti-asialo GM1 was administrated every 2–3 days or 4–5 days respectively, until day 12. Animals in antibody treatment groups were maintained on a normal (drug-free) diet, while a control group received TAC diet (pre-supplied for more than one week before SCC challenge) to confirm the viability of transferred SCC cells (TAC group).

To examine the influence of immune cell subsets on SCC regression, mice were maintained on TAC diet following SCC challenge for 9 days to allow SCC growth. On day 9 post SCC challenge, TAC diet was removed (except for the mice in the control TAC group) and antibody treatments were administrated according to each treatment-group. Antibody injections were initiated on the same day as TAC diet removal, and then repeated on day 16 for anti-CD8β/ anti-TCRβ/ appropriate isotype; days 16 and 23 for anti-CD4/ anti-TCRγδ; every 2–3 days for anti-Gr-1 (until day 23); and every 4–5 days for anti-asialo GM1 (until day 23).

### 2.7. Schedule of Anti-IFN-γ and Anti-CXCR3 Treatments

IFN-γ neutralisation, and CXCR3 blocking, was initiated when SCC tumours were well-established (day 15 post SCC challenge). Mice were injected with anti-IFN-γ or anti-CXCR3 or appropriate isotype mAb every two days (starting on day 15). On day 16, TAC diet was removed. A control group remained on TAC to confirm SCC viability. The short-term anti-IFN-γ treatment regime continued to day 29 while the long-term treatment regime continued until day 77. Anti-CXCR3 injections continued until day 51.

### 2.8. Cell Isolation from Blood, Lymph Node, and Tumour

Approximately 100 µl of blood was added to 10 mL ACK lysis buffer (150 mM NH_4_Cl, 10 mM KHCO_3_, 0.1 mM EDTA, pH 7.2–7.4) and centrifuged at 340 g for 10 min at 4 °C. The cell pellet was resuspended and incubated in a further 3 mL of ACK lysis buffer for 3 min. Samples were topped up with an additional 3 mL FACS buffer (2% fetal bovine serum (FBS, Thermo Fisher Scientific) in PBS) and centrifuged at 340 g for 6 min at 4 °C. Processed cells were then washed once with fresh FACS buffer and transferred into a 96-well-plate or FACS tubes for staining.

For the isolation of lymphocytes from lymph nodes, harvested lymph nodes were gently pressed through a 70 μm cell strainer using a syringe plunger. To release cells from tumours, harvested tissue was cut into small fragments and digested for 70 min at 37 °C in RPMI media containing 2% FBS, 3 mg/mL collagenase D and 5 ug/mL DNase I. Tissues were then gently pressed through a 70 μm cell strainer to create a single-cell suspension.

### 2.9. Phenotypic Analysis of Cell Populations by Flow Cytometry

Isolated cells were resuspended in FACS buffer and incubated with Fc-block (Purified Rat Anti-Mouse CD16/CD32: isotype Rat IgG2b, clone: 2.4G2, BD Biosciences) for 20 min on ice to block non-specific antibody staining. Monoclonal antibodies for surface staining were subsequently added and incubated on ice for 30–40 min in concert with Live/Dead Aqua Stain (Thermo Fisher Scientific) to elucidate live cell populations. Intracellular staining for CXCR3 expression was performed using the Foxp3 Transcription Factor Staining Buffer Set Kit (eBioscience) as per the manufacturers’ instructions. Immediately before FACS acquisition, Flow-Count^TM^ Fluorospheres (Beckman Coulter, Miami, FL, USA) were added to allow assessment of total cell counts. The antibodies used for flow cytometry are listed in Appendix A. Flow cytometric analysis were performed using LSR Fortessa (BD Biosciences) flow cytometers with FACSDiva software (Becton Dickinson, Sparks, MD, USA). Data were exported and analyzed using FlowJo software (Treestar Inc., Ashland, OR, USA).

### 2.10. Chemokine Analysis

Tumours were harvested in cold PBS and cut into fine pieces, which were randomized for flow cytometry analysis (described above) or chemokine analysis. For chemokine analysis, excised specimens were weighed and mechanically homogenized in 500 µL of PBS containing protease inhibitor cocktail (cOmplete^TM^ ULTRA Tablets, Mini, EDTA-free, EASYpack; Roche; Indianapolis, IN, USA), then frozen at −80 °C. Homogenates were further utilized to evaluate the presence of 12 different chemokines using the LEGENDplex^TM^ Mouse Proinflammatory Chemokine Panel (Biolegend) according to the manufacturer’s instructions. This analysis was performed on a CytoFLEX S Flow Cytometer (Beckman Coulter, Lane Cove, Australia) and acquired data were analyzed using LEGENDplex^TM^ Data Analysis Software (Biolegend). All measurements were normalized to starting lesion weight (pg per gram wet tissue weight).

### 2.11. Chimeric Mouse Model

A chimeric mouse model was designed to permit experimental assessments of C57BL/6 gene knockout (KO) mouse immune cells in conjunction with the FVB SCC cell line. HPV38E6E7 FVB x C57BL/6 F1 progeny (CD45.1^+^ and CD45.2^+^) were irradiated with 1000 cGy (two 500 cGy doses with an interval of 4 h). Irradiated F1 mice were reconstituted by retro-orbital injection of 5 million bone marrow (BM) cells from T cell-depleted donor mice. To assess bone marrow transplantation engraftment rates, blood samples were stained with anti-CD45.1 and -CD45.2 specific mAbs and analysed by flow cytometry. Bone marrow engraftment was determined by the ratio of CD45.2^+^ CD45.1^−^ / total CD45^+^ (>8 weeks after injection). Mice with more than 80% engraftment were used for subsequent experiments. To investigate the anti-tumour immunity of chimeric mice, mice were challenged with SCC tumour cells as described above.

To investigate the importance of IFN-γ secretion by CD8^+^ T cells for the rejection of SCC tumours, CD8^+^ T cells from wild-type chimeras that had controlled tumour growth were transferred into tumour-bearing IFN-γ^−/−^ chimeras. Spleens and lymph nodes were harvested from wild-type chimeras that had rejected 2 rounds of SCC challenge with SCC cells. Harvested cells were negatively purified for CD8^+^ T cells using a bead enrichment kit (EasySep™ Mouse CD8^+^ T Cell Isolation Kit (Stemcell Technologies, Vancouver, BC, Canada)), and the purity of these cells was examined by FACS analysis. Purified CD8^+^ T cells (5 × 10^6^/mouse) were transferred into recipients with growing tumours that were in the range of 0.1–0.3 cm^3^.

### 2.12. Statistical Analysis

Unless otherwise stated, statistical analysis was carried out using GraphPad Prism version 7.03 (GraphPad Software, San Diego, CA, USA). Statistical tests are as indicated in figure legends. The correlations of chemokine abundance with T cell infiltration were analysed using a two-tailed Spearman correlation analysis. Relationship strengths were described as follows: (r) > 0.5 = strong; 0.3–0.5 = moderate; <0.3 = weak. A *p* value of *p* < 0.05 (*) was considered significant. *p* < 0.01 (**), *p* < 0.001 (***), and *p* < 0.0001 (****) are indicated. The heatmap of Spearman correlation r values and radar chart of mean log-abundance values of chemokines were plotted by using the R package, as previously described [28].

## 3. Results

### 3.1. CD8^+^ T Cells Are Critical for the Control of SCC

To determine immune cell types critical for preventing SCC tumour establishment and mediating SCC tumour rejection, we established a transplantable SCC cell line in our laboratory by UV-treatment of the skin of an HPV38E6E7-FVB transgenic mouse. These mice express Human Papilloma Virus (HPV) 38 E6 and E7 proteins in their keratinocytes, and reproducibly form SCC when subjected to UV three times per week for 25 weeks [25]. The SCC cell line reproducibly forms tumours when injected subcutaneously into mice administered tacrolimus in the diet (TAC; Figure 1A) but does not form tumours in immune-competent mice (e.g., Isotype-antibody treated, Appendix A).

To uncover immune populations necessary to prevent SCC tumour establishment, a series of immune cell depletions were performed in otherwise naïve mice before SCC cell line challenge (Appendix A). Depletion of CD8β^+^ T cells or TCRβ^+^ T cells was sufficient to permit SCC tumour establishment (Appendix A). In contrast, mice depleted of CD4^+^ T cells, TCRγδ^+^ T cells, NK cells, or Gr-1^+^ myeloid cells, remained capable of preventing SCC tumour establishment (Appendix A). Approximately four weeks after cell line challenge, tumours continued to grow in the majority of mice depleted of CD8β^+^ T cells or TCRβ^+^ T cells, but not in mice depleted of CD4^+^ T cells (Appendix A), suggesting that CD8^+^ T cells also play a vital role in the prevention of SCC tumour progression.

To determine immune populations critical for mediating the regression of established tumours, we took advantage of our SCC ‘regressor’ model, in which SCC tumours established in mice fed with TAC-diet undergo regression following the cessation of TAC-diet administration (Figure 1B). On day 9 post tumour challenge, at the same time as TAC-diet withdrawal, specific immune cell populations were individually depleted (Figure 1C). Depletion of CD8β^+^ T cells or TCRβ^+^ T cells resulted in an inability to reject established SCC tumours (Figure 1D). In contrast, treatment with an isotype-control antibody, or depletion of CD4^+^ T cells, TCRγδ^+^ T cells, NK cells, or Gr-1^+^ myeloid cells, did not prevent SCC tumour rejection (Figure 1D). Tumours grew progressively in mice lacking either CD8β^+^ T cells or TCRβ^+^ T cells, resulting in reduced survival (Figure 1E). Together, the data demonstrate a key role for CD8^+^ T cells in the control of SCC establishment, progression, and regression.

### 3.2. CD4^+^ T Cells Contribute to the Control of SCC Growth

The disparity in SCC growth rate and consequently mouse survival when comparing TCRβ^+^ T cell depletion to CD8β^+^ T cell depletion (Figure 1) suggests a role for CD4^+^ T cells and possibly other immune subsets in combatting the rate of SCC growth. In a parallel set of experiments to those conducted in Figure 1, at the same time as TAC-diet withdrawal, mice with established SCC were injected with CD8β^+^ T cell-depleting antibody alone, or CD8β^+^ T cell-depleting antibody in combination with other subset-depleting antibodies (Figure 2A,B). In the absence of CD8^+^ T cells, the majority of mice were unable to mediate SCC rejection, as expected. When comparing the effect of depleting CD4^+^ T cells, or TCRβ^+^ T cells, or TCRγδ^+^ T cells, or NK cells, or Gr-1^+^ myeloid cells in combination with CD8β^+^ T cells, three growth phenotypes emerged: (1) tumours grew faster (Figure 2B, left hand panels), or (2) tumours grew at a similar rate (Figure 2B, middle panels), or (3) tumours grew slower (Figure 2B, right hand panels). Combining CD8β^+^-depletion with either CD4^+^-depletion or TCRβ-depletion resulted in a faster rate of SCC growth and significantly reduced survival when compared to CD8β^+^-depletion alone (Figure 2B, left hand panels). Notably, there were no significant differences in survival between ‘Anti-CD8β + Anti-CD4’ and ‘Anti-CD8β + Anti-TCRβ’ groups. Together, the data indicate a role for CD4^+^ T cells in controlling the rate of SCC growth. However, combining TCRγδ^+^ T cell or NK cell-depletion with CD8β+ T cell depletion did not show a significant effect on tumour growth or survival (Figure 2B, middle panels). Interestingly, a slow-down in tumour growth (Figure 2B, top right panel) and improved survival (Figure 2B, bottom right panel) was apparent when ‘Anti-CD8β + Anti-Gr-1’ was compared to CD8β^+^-depletion alone, and two out of eight animals could reject established SCC tumours. Remarkably, the data indicate that the depletion of Gr-1^+^ myeloid cells allowed some mice to control tumour growth in the absence of CD8^+^ T cells, although ultimately, the majority of mice lacking CD8^+^ T cells were unable to reject established tumours. Taken together, the data suggest a positive role for CD4^+^ T cells and a negative role for Gr-1^+^ myeloid in the control of SCC growth.

IFN-γ is a well-known Th1 effector cytokine. To define whether CD4^+^ T cells, and CD8^+^ T cells, produce IFN-γ during SCC regression, mice were fed TAC-diet and challenged with SCC cells. Fourteen days later TAC diet was withdrawn, and tumours and draining lymph nodes were harvested after a further seven days (day 21; when tumours were a similar size to control animals) or after a further 12 days (day 26; when tumours were visibly regressing, Figure 2C) and analysed for IFN-γ production by FACS. As shown in Figure 2D, following TAC removal there was a significant increase in the number of CD8^+^ T cells in the draining lymph nodes producing IFN-γ at day 21 but not by day 26. Conversely, in tumours, there was a variable but significant increase in IFN-γ-production by CD8^+^ T cells at day 21, and a strong and consistent production of IFN-γ by CD8^+^ T cells at day 26. CD4^+^ T cells, in contrast, did not produce detectible IFN-γ in the draining lymph nodes at either time point, but displayed a small but significant increase in the number of cells producing IFN-γ within the tumour mass at both time points. The results indicate that both CD8^+^ T cells and CD4^+^ T cells produce IFN-γ in the tumour environment following the removal of TAC.

### 3.3. Neutralising IFN-γ Prevents SCC Regression

To investigate a role for IFN-γ in the control of SCC, mice with established SCC tumours were injected with an anti-IFN-γ neutralising antibody or an isotype control antibody at the same time as TAC-diet withdrawal (Figure 3A). An additional group established in parallel remained on TAC-diet throughout the experiment and served as a tumour growth control (Figure 3B, left panel). Tumours grew consistently and progressively in these mice. Isotype antibody-treated control mice rejected SCC tumours following the removal of TAC-diet (Figure 3B, central panel), as expected. However, SCC continued to grow progressively in mice that received neutralising anti-IFN-γ antibody over a period of 14 days (short-term; Figure 3B, right panel), but the majority regressed once the antibody administration was stopped (Figure 3B,C). To confirm the long-term importance of IFN-γ for the control of tumour growth, we extended the anti-IFN-γ antibody treatment period from 14 days to 62 days (long-term; Figure 3D). Notably, following long-term anti-IFN-γ treatment, the majority of tumours grew progressively (Figure 3D), resulting in significantly reduced survival compared to untreated control or mice receiving short-term anti-IFN-γ treatment (Figure 3E). Together, the data indicate a significant role for IFN-γ in SCC tumour regression.

### 3.4. IFN-γ Is Not Directly Cytotoxic to SCC Cells

To determine whether IFN-γ is directly cytotoxic to SCC cells, we cultured the SCC cell line with recombinant mouse IFN-γ (10 ng/mL or 100 ng/mL) in vitro for 72 h and then assessed cell death by FACS. As shown in Figure 4A–C, IFN-γ treatment did not impact upon the proportion of live (Annexin V^−^ 7-AAD^−^), apoptotic (Annexin V^+^ 7-AAD^−^), or necrotic (Annexin V^−^ 7-AAD^+^) cells. However, IFN-γ treatment induced the upregulation of MHC class I expression (Figure 4D) and increased the secretion of CXCL10 (Figure 4E), which was preventable in both cases through the addition of a neutralising anti-IFN-γ antibody during co-culture (Figure 4D,E). Interestingly, IFN-γ treatment did not induce detectable CXCL11 expression (Figure 4F). Together, the data indicate that while IFN-γ is not directly cytotoxic to SCC cells and does not induce growth arrest, IFN-γ has biological effects on SCC cells that may enhance T cell killing.

### 3.5. IFN-γ Neutralisation Impairs the Infiltration of CXCR3^+^ CD8^+^ T Cells Into SCC

Having established that IFN-γ increases MHC Class I expression and induces CXCL10 secretion from SCC cells in vitro, we next examined whether IFN-γ may play a role in modulating MHC Class I expression in vivo and initiating the infiltration of T cells into tumours. Mice with established tumours were treated with an anti-IFN-γ neutralising antibody around the same time as TAC-diet withdrawal (as in Figure 3), and tumours and draining inguinal lymph nodes (iLN) were subsequently harvested during the treatment window and analysed for MHC Class I expression (tumours) and T cell abundance (both tumours and iLN) by FACS (Figure 5A). Neutralising IFN-γ antibody treatment did not result in statistically reduced MHC I expression on CD45^−^ cells within tumours (Appendix A). Furthermore, as shown in Figure 5B, the abundance of CD4^+^, CD8^+^, and γδ T cell subsets in the iLN did not differ significantly in response to anti-IFN-γ treatment. In contrast, anti-IFN-γ dramatically reduced the infiltration of CD4^+^- and CD8^+^ T cells into tumours (Figure 5C). To determine whether the neutralisation of IFN-γ might impact upon the CXCR3-CXCL9/10/11 chemokine axis, we examined the abundance of CXCR3^+^ T cell subsets in tumours and iLN. In the lymph nodes, the abundance of CXCR3^+^ T cells did not differ significantly in response to anti-IFN-γ treatment (Figure 5D). However, anti-IFN-γ significantly impaired the infiltration of CD8^+^CXCR3^+^ and CD4^+^CXCR3^+^ T cell subsets into the tumour mass (Figure 5E). The results indicate that IFN-γ plays a direct or indirect role in the recruitment of CD8^+^CXCR3^+^ and CD4^+^CXCR3^+^ T cell subsets into SCC tumours.

### 3.6. IFN-γ Neutralisation Diminishes CXCL10 and CCL5 Production Within SCC

To explore how the tumour environment is influenced by IFN-γ, we examined the expression level of 12 chemokines known for their capacity to induce immune cell migration into tumours. Mice with established tumours were treated with an anti-IFN-γ neutralising antibody at the same time as TAC-diet withdrawal (as in Figure 5A). Tumours were harvested and examined for T cell content by FACS and chemokine content by cytometric bead array (Figure 6). Anti-IFN-γ treatment significantly impaired the production of CXCL9, CXCL10. and CCL5 when compared to isotype-treated tumours (Appendix A), but had no impact on CXCR3 expression by T cell subsets (Appendix A). In contrast, the production of CCL2, CCL17 and CCL22 was significantly increased following anti-IFN-γ treatment.

To determine how T cell infiltrates correlate with chemokine expression, we performed a correlation analysis in anti-IFN-γ/isotype treated tumours. As shown in Figure 6B and Appendix A, strong positive-correlations (r > 0.5, *p* < 0.05) were observed between the expression of CXCL10 and CCL5 and CD8^+^ T cell abundance in tumours, while CCL17 and CCL22 were negatively correlated (r < −0.5, *p* < 0.05) with CD8^+^ T cell infiltration. CD4^+^ T cell infiltration was positively correlated with CCL5 and CXCL13 expression (r > 0.5, *p* < 0.05). Correlations were weak (−0.3 < r < 0.3) or not statistically-significant between CCL2, CCL3, CCL4, CCL11, CCL20, CXCL1, CXCL9, CXCL13, and T cell infiltrates. In summary, we observed that when IFN-γ was neutralised, the production of CXCL10 and CCL5 within SCC tumours was notably impaired, which correlated with a significant reduction in infiltrating CD8^+^ T cell numbers.

### 3.7. CXCR3-Blockade Prevents CD8^+^ T Cell Recruitment and SCC Regression

Next, we examined blockade of CXCR3 to confirm a role for the CXCR3/CXCL10/11 chemokine axis in CD8^+^ T cell-mediated SCC regression. We firstly determined that SCC cells were CXCR3^−^ (Appendix A). Mice with established tumours were treated with an anti-CXCR3-blocking antibody at the same time as TAC-diet withdrawal (Figure 7A). Notably, the anti-CXCR3-blocking antibody (clone CXCR3-173) used in these experiments has been reported previously to block the binding of CXCL10 and CXCL11 to CXCR3, but not the binding of CXCL9 to CXCR3 [29]. As shown in Figure 7B (right hand panel), treatment with anti-CXCR3 antibody prevented tumour regression, and led to significantly reduced survival (Figure 7C). To investigate the impact of CXCR3 blockade on the abundance of T cell subsets within iLN and tumours, a parallel experiment was set up in which iLN and tumours were harvested on day 28 (when isotype-antibody control tumours had started to reject). In the lymph nodes, the abundance of CD4^+^, CD8^+^, and γδ T cell subsets was similar between anti-CXCR3 treated- and isotype-treated mice (Figure 7D). However, CXCR3 blockade significantly reduced CD8^+^ T cell infiltration, but not CD4^+^- or γδ T cell infiltration, into tumours (Figure 7D). These results highlight the importance of the CXCR3/CXCL10/11 chemokine axis in CD8^+^ T cell-mediated SCC tumour regression and suggest that CXCR3 blockade prevents SCC regression by impairing the infiltration of CD8^+^ T cells into tumours.

### 3.8. IFN-γ Secretion by Immune Cells Is Important for SCC Control

To explore whether immune cells themselves are a source of the IFN-γ needed to control SCC growth, we designed a chimeric mouse model to permit experimental assessments with our FVB SCC model using available C57BL/6 gene knock-out mouse strains. In this model, bone marrow cells collected from T cell-depleted C57BL/6 (wild-type; WT), IFN-γ^−/−^ or IFN-γR^−/−^ mice were transferred into pre-irradiated HPV38E6E7-FVB × C57BL/6 F1 progeny. Unlike FVB mice, WT chimeric animals did not show a consistent ability to reject established tumours when TAC-diet was withdrawn (data not shown). Therefore, we examined the ability of unmanipulated chimeric mice to prevent SCC establishment (i.e., the mice did not receive TAC-diet at any stage). As shown in Figure 8A, SCC cells injected into WT chimeras failed to establish in 14 out of 15 mice, indicating that the majority of WT chimeras were able to control SCC outgrowth to some extent. In IFN-γ^−/−^ chimeras, cells of the hematopoietic lineage were unable to produce IFN-γ, however stroma/SCC cells could. Only 4 out of 10 IFN-γ^−/−^ chimeras were able to prevent SCC tumour growth, leading to reduced overall survival (Figure 8A; *p* = 0.006, Fisher’s exact test, and Figure 8B) despite having a similar bone marrow engraftment rate (Figure 8C). Thus, IFN-γ produced by hematopoietic cells appears to contribute to the control of SCC. Interestingly, nearly all IFN-γR^−/−^ chimeras lacking IFN-γ-receptor expression on hematopoietic cells were able to prevent the growth of SCC (Figure 8A), suggesting that while IFN-γ produced by hematopoietic cells is important for tumour control, IFN-γ-signalling to hematopoietic cells is not.

Finally, we investigated whether IFN-γ produced from CD8^+^ T cells specifically, is important for the control of established SCC. We transferred CD8^+^ T cells from WT chimeras that had prevented SCC establishment, into IFN-γ^−/−^ chimeras bearing established SCC tumours (Figure 8D). IFN-γ^+^ CD8^+^ T cell transfer resulted in tumour growth arrest (Figure 8E) and increased survival (Figure 8F). These results suggest that IFN-γ production by CD8^+^ T cells is important for the control of SCC growth.

## 4. Discussion

Recently, we conducted a study of the cytokine and chemokine environment within advancing stages of the SCC disease spectrum in patients [7]. Samples taken from Actinic Keratosis, considered to be a premalignant or precancerous form of SCC associated with a high rate of spontaneous regression, indicated significant increases in several proinflammatory cytokines (e.g., IFN-γ, TNF-α, IL-1β, IL-17A, IFN-α, IL-12p70) when compared to photodamaged skin at the protein level. Of these, the largest increase was in IFN-γ, which corresponded with an increase in the chemokine CXCL10. These results prompted us to try and define the role that IFN-γ may play in the control of SCC.

The growth of tumours in otherwise immune-competent humans and mouse models poses considerable challenges when trying to ascertain how the immune system would normally prevent SCC emergence and/or induce SCC rejection. In the present study therefore, we developed an SCC ‘regressor’ model. We developed a cell line from an SCC tumour that arose following the chronic UV-treatment (25 weeks) of mice engineered to express the Human Papilloma Virus 38 E6 and E7 oncogenes under the control of the K14 promoter [25]. Unlike other HPV transgenic mice, which often form multiple tumour types spontaneously [30,31,32], or non-transgenic mice strains, which tend to favour the formation of other types of tumours such as fibrosarcoma following UV-irradiation [33,34], HPV38E6E7^+^ mice consistently form SCC tumours in response to chronic UV irradiation [25]. When transplanted into naïve immunocompetent mice the SCC cell line failed to establish tumours, in agreeance with other models of UV-induced tumour transfer described previously [13,14,15]. Depletion of immune subsets prior to transfer allowed us to investigate the role of specific cell types in the prevention of SCC establishment. Furthermore, transient immunosuppression using tacrolimus, a drug currently the mainstay of treatment regimens to prevent solid organ rejection in transplant patients [35], facilitated the establishment of SCC tumours following SCC cell line transfer. Subsequently, the cessation of tacrolimus treatment allowed us to conduct an examination of the role played by immune subtypes and IFN-γ in SCC tumour rejection.

CD8β antibody-depletion allowed both tumour establishment in naïve mice, and the progressive growth of SCC following tacrolimus removal in our rejection model, highlighting a critical role for CD8^+^ T cells in the control of SCC. In the rejection model, our data further indicated that survival time is significantly reduced when mice are depleted of TCRβ^+^ cells in general, evidencing a role for CD4^+^ T cells in the anti-tumour response. Here, the role of CD4^+^ T cells may be through support for CD8^+^ T cell activation as an early source of IL-2 [36], cooperation with CD8^+^ T cells during the effector phase to kill tumour stroma more effectively [37], and/or as an important source of IFN-γ production within the tumour microenvironment (Figure 2D and [38]). However, given that mice depleted of CD4^+^ cell populations are able to reject established SCC, the role of CD4^+^ T cells in SCC rejection appears to be additive as opposed to essential.

Other groups have indicated a role for IFN-γ in the immunosurveillance of 3-Methylcholanthrene-induced skin tumours following the observation that IFN-γ-deficient mice are more susceptible to skin tumour formation than wild-type mice [22,23]. In our model of UV-induced SCC, mice were unable to control the growth of established tumours while receiving an antibody that neutralised IFN-γ. When treatment with the neutralising antibody ceased, however, the majority of mice were able to completely reject- or substantially reduce the volume of- their established tumours. The data indicate a critical role for IFN-γ in the control of UV-induced SCC. However, IFN-γ was found not to be directly cytotoxic to SCC cells in vitro.

Although links between MHC I expression and tumour control are well-documented [39], mice treated with an IFN-γ neutralising antibody did not show statistically reduced MHC I expression on CD45^−^ cells within their tumours. Our data indicate, however, that antibody-treated mice did show reduced CXCL9, CXCL10, and CCL5 chemokine production within the tumour mass, and a corresponding reduction in the number of tumour-infiltrating T cells. The CXCR3-ligands CXCL9, CXCL10, and CXCL11 all increase in abundance with advancing SCC disease stage in humans [7]. In our model, mice treated with an antibody to block the binding of CXCL10 and CXCL11 to CXCR3 (but not the binding of CXCL9 to CXCR3 [29]) were unable to control the growth of established tumours. Treatment was shown to result in a significant reduction in CD8 T cell infiltration into tumours, but not CD4 T cell or γδ T cell infiltration, which corresponds with our earlier observation of the key role played by CD8 T cells in SCC rejection. These results indicate a key role for the CXCL10/CXCL11/CXCR3 axis in SCC rejection. Although IFN-γ induced CXCL10 secretion from SCC cells in vitro, we could not detect a similar induction of CXCL11 secretion following IFN-γ treatment. However, the possibility remains that in vivo CXCL11 is produced by other cells present within the tumour microenvironment.

Using our chimeric models, we have established the importance of IFN-γ produced by hematopoietic cells in the control of SCC, and furthermore, that IFN-γ produced by CD8^+^ T cells plays a role. In other experiments we have also established that CXCL10 production is key for CD8 T cell-mediated SCC rejection. While it is feasible to argue that IFN-γ produced by CD8^+^ T cells within the tumour acts in a positive feedback loop to induce further CXCL10 expression from SCC cells, leading to further CD8^+^ T cell infiltration, this loop does not explain the initiating, hematopoietic, source of IFN-γ within the tumour microenvironment that led to CXCL10 induction in the first place. Many innate immune cells are known to release IFN-γ, including NK cells, NKT cells, γδ T cells, dendritic cells and macrophages [24], and co-culture experiments in the laboratory have confirmed that SCC cells can directly stimulate IFN-γ release by NK cells [40]. Therefore, it is possible that innate immune cells, such as NK cells, provide an early source for IFN-γ. Although it is interesting to note that the depletion of NK cells (or γδ T cells or Gr-1^+^ monocytes/macrophages/dendritic cells) cells did not prevent tumour regression in our model (Figure 1D), it is possible that IFN-γ had already been released by these cells into the tumour microenvironment prior to their depletion on day 9. Our data highlight a strong association between IFN-γ production and the infiltration of CD8 T cells and CXCR3^+^ CD8 T cells in particular, into SCC. IFN-γ induces both CXCL10 and CCL5 [41,42], and both chemokines showed a strong positive correlation with CD8 T cell infiltration into tumours. The observation that an IFN-γ neutralising antibody can block CXCR3^+^CD4^+^ T cell infiltration into the tumour mass (Figure 5E), but an antibody that blocks the binding of CXCL10 and CXCL11 to CXCR3 cannot (Figure 7D, suggests that while CXCL10 is important for the infiltration of CD8^+^ T cells into SCC, different chemokines may influence the migration of CD4^+^ T cells into SCC. In addition to CXCL10, the injection of anti-IFN-γ also significantly impaired the production of CXCL9 and CCL5 in the tumour microenvironment compared to isotype antibody-treated tumours (Appendix A). Our data also indicate a strong positive correlation between CCL5 and CD4^+^ T cell abundance in tumours (Figure 6B and Appendix A). Therefore, IFN-γ also appears to play an important role in the infiltration of CD4^+^ T cells into SCC, possibly through the induction of CCL5. In esophageal SCC, CXCL10 and CCL5 have been positively correlated with CD8 and Granzyme B at the mRNA level, and tissue expression of CCL5 was positively associated with post-surgical patient survival [43]. CCL5 is known to be increased in SCC compared to photodamaged skin, actinic keratosis, and intraepithelial carcinoma [7], but interestingly does not increase significantly when photodamaged skin is compared to actinic keratosis and intraepithelial carcinoma [7], suggesting that the expression of this chemokine is a relatively late event during SCC progression in humans [7]. Although we have not looked at an absolute requirement for CCL5 expression in the control of SCC in the present study, it is interesting to speculate that CCL5 may also be responsible for the small but detectible numbers of CD8 T cells present within the tumours of anti-CXCR3-treated mice. Further characterization of CCL5 is warranted to extend our understanding of its role in SCC regression.

## 5. Conclusions

In conclusion, our studies investigated the role of IFN-γ in the control of established SCC tumours in a UV-induced SCC rejection model and showed that IFN-γ plays a critical role in the induction of chemokines that permit the infiltration into- and destruction of- SCC by CXCR3^+^ CD8 T cells. Our results advocate further study into immune-mediated SCC rejection pathways, and factors that may disrupt their integrity, in patients.

## Figures and Tables

**Figure 1 cancers-13-02131-f001:**
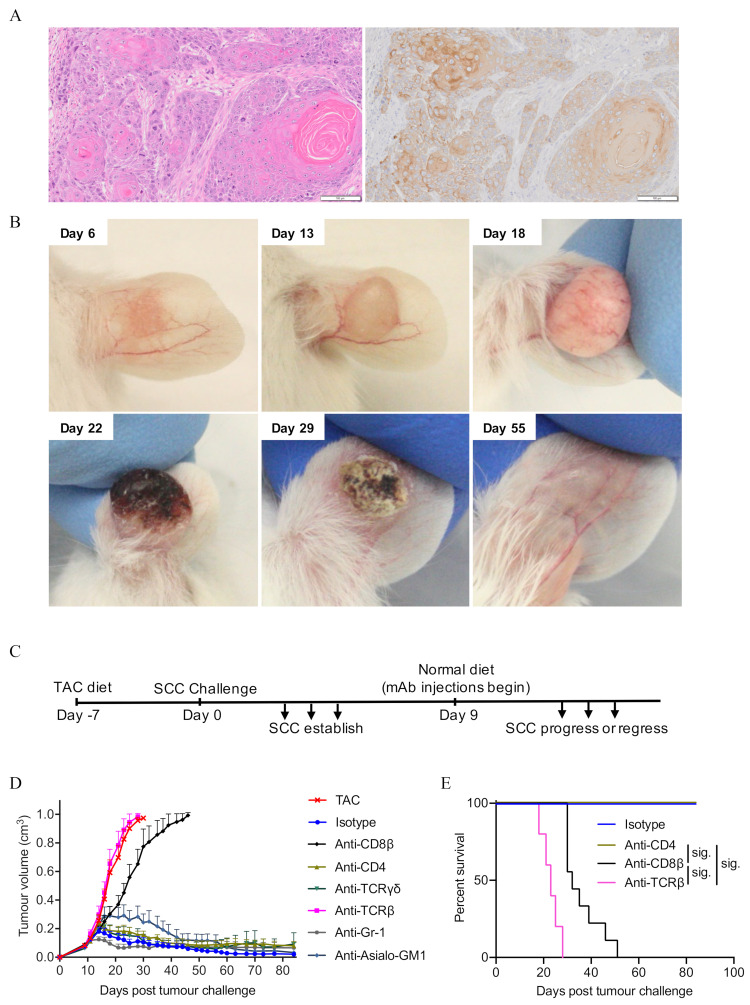
Depletion of immune subsets highlights a role for CD8^+^ T cells as mediators of SCC tumour regression. (**A**) Representative photomicrographs of SCC tumours growing in immunosuppressed mice. Left hand side: H&E; Right hand side: pan-cytokeratin. Scale bar = 100 μm. (**B**) Example of established SCC regression following the removal of tacrolimus diet on day 13. Ear model used for illustrative purposes only. (**C**) Treatment schedule for SCC regression experiments. (**D**) SCC regression following depletion of immune subsets with the indicated antibodies. Combined data from 3 independent experiments (mean ± SEM) shown, n = 5–13/group. (**E**) Mouse survival in (**D**). Survival analyses was performed by Log-rank (Mantel-Cox) test. Results then displayed as significant (sig.) after bonferroni-corrected multiple comparison test. Bonferroni-corrected threshold = 0.0125, K = 4.

**Figure 2 cancers-13-02131-f002:**
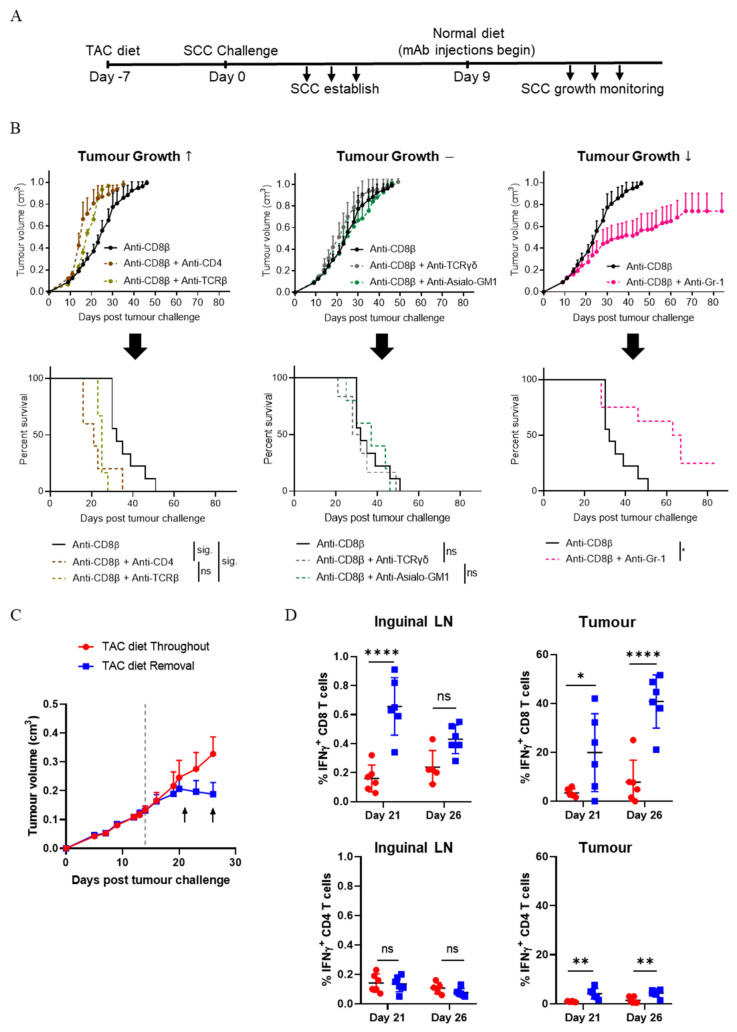
Multiple immune subsets impact upon SCC growth rate. (**A**) Treatment schedule for immune subset depletion experiments. (**B**) Tumour growth rate and corresponding survival curves following depletion with the indicated antibodies. Combined data from 3 independent experiments (mean ± SEM) shown, n = 5–13/group. Survival analyses was performed by Log-rank (Mantel-Cox) test. Results then displayed as significant (sig.) or not significant (ns) after bonferroni-corrected multiple comparison test (bottom left and bottom middle). Bonferroni-corrected threshold = 0.016, K = 3. Bottom right: Survival analyses was performed by Log-rank (Mantel-Cox) test (*, *p* < 0.05; **, *p*<0.01; ****, *p*<0.0001). (**C**) Time points assessed for T cell-associated IFN-γ production in lymph nodes and tumours of mice undergoing SCC regression. TAC diet removed on day 14 as indicated by horizontal dotted line. Arrows indicate harvest timepoints (left, day 21; right, day 26). Mean ± SEM, n = 12/group. (**D**) Analysis of day 21 and day 26 CD8^+^-and CD4^+^ T cell IFN-γ production as described in (**C**). n = 6/time point/group. Two-way ANOVA, bars represent mean ± SEM, data points represent individual mice.

**Figure 3 cancers-13-02131-f003:**
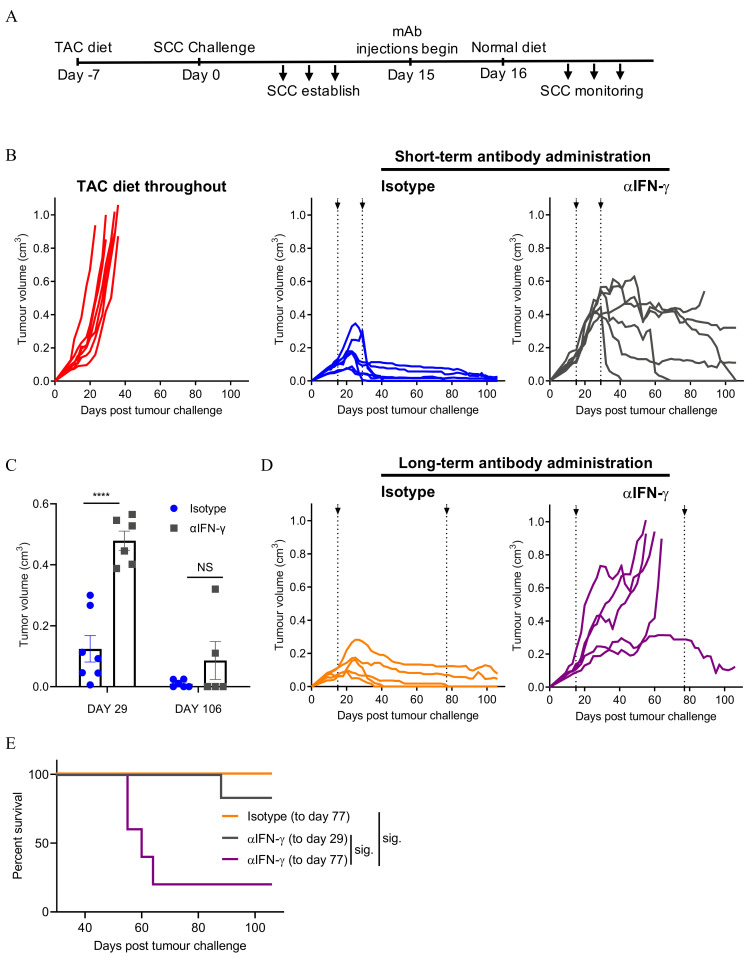
IFN-γ neutralisation abrogates SCC rejection. (**A**) Experimental schedule. (**B**) SCC growth pattern following short-term treatment with the indicated antibodies. Combined data from 2 independent experiments, individual tumour growth shown, n = 6–7/group. (**C**) Tumour volume during (Day 29) and after cessation (Day 106) of anti-IFN-γ treatment. Two-way ANOVA, bars represent mean ± SEM, data points represent individual mice. *p* < 0.0001 (****), (**D**) SCC growth pattern following long-term treatment with the indicated antibodies. Combined data from 2 independent experiments, individual tumours shown, n = 5/group. (**E**) Comparison of mouse survival in B and D. Survival analyses was performed by Log-rank (Mantel-Cox) test. Results then displayed as significant (sig.) after bonferroni-corrected multiple comparison test. Bonferroni-corrected threshold = 0.016, K = 3. Dotted lines in (**B**,**D**) indicate the period of antibody administration.

**Figure 4 cancers-13-02131-f004:**
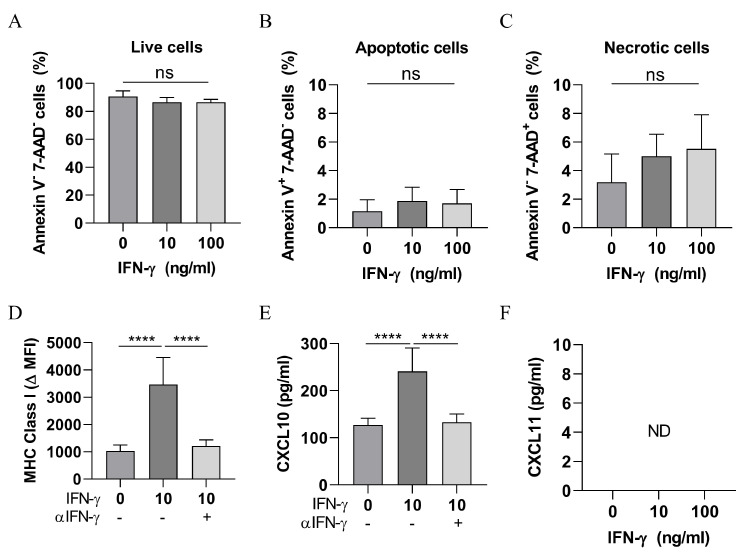
Direct effects of IFN-γ on SCC. (**A**–**C**) Viability of SCC cells following IFN-γ treatment. SCC cells were treated with IFN-γ for 72 h and then assessed for Annexin V and 7AAD staining by flow cytometry. Bars represent mean ± SD, n = 6/group, one-way ANOVA. (**D**) MHC class I expression. The delta mean-fluorescence-intensity (∆MFI) of H-2K^q^ is shown. Bars represent mean ± SD, n = 6–9/group, one-way ANOVA followed by Tukey’s multiple comparisons test. *p* < 0.0001 (****), (**E**) CXCL10 and (**F**) CXCL11 secretion in supernatants assessed by ELISA assay. Bars represent the mean ± SD, n = 6–9/group, one-way ANOVA followed by Tukey’s multiple comparisons test (**E**). Data are derived from 3 independent experiments with similar results. *p* < 0.0001 (****), ND = not detected. ns = not significant.

**Figure 5 cancers-13-02131-f005:**
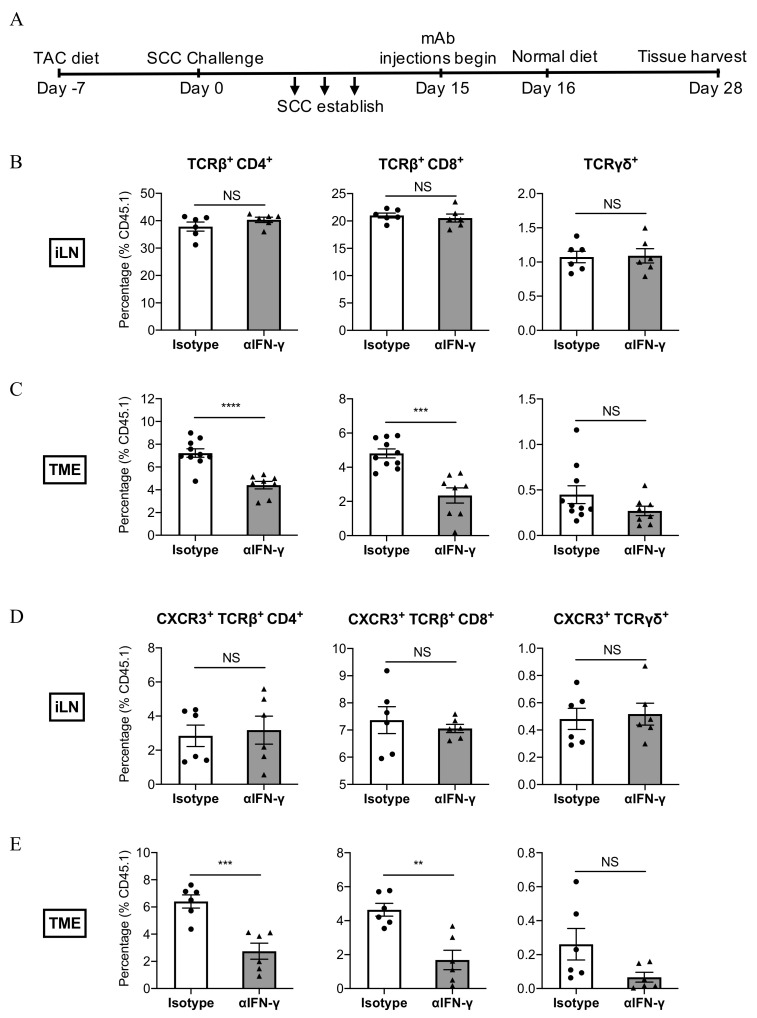
IFN-γ neutralisation reduces the number of CD8^+^ T cells that infiltrate into SCC. (**A**) Experimental schedule. (**B**,**C**) Abundance of T cell subsets, and (**D**,**E**) abundance of CXCR3-expressing T cell subsets in the inguinal lymph nodes (iLN) or tumour microenvironment (TME) respectively. Indicated antibody treatments started on day 15 and continued up to the point of tissue harvest. Each dot corresponds to one mouse, bars represent the mean ± SEM. (**B**–**E**) Data are derived from 3 independent experiments, n = 6–10/group, and are normalised to CD45.1 expression to mitigate the impact caused by large differences in total cell counts when comparing mice with regressing and non-regressing tumours. Statistical analysis was performed by unpaired Student’s *t*-test. *p* < 0.01 (**), *p* < 0.001 (***), *p* < 0.0001 (****).

**Figure 6 cancers-13-02131-f006:**
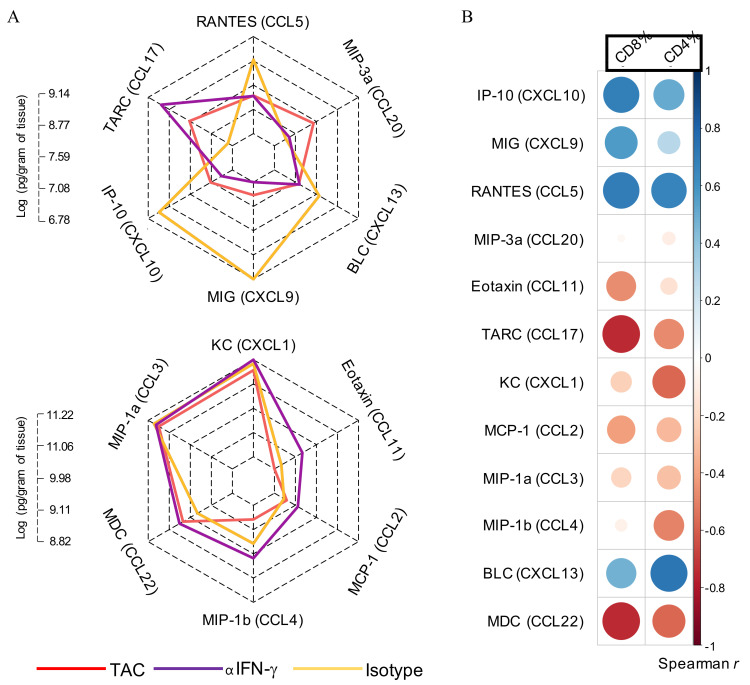
IFN-γ neutralisation alters chemokine abundance within the tumour microenvironment. (**A**) Homogenized tumour samples were analyzed by cytometric bead array to determine chemokine content. Radar charts of mean log-abundance values of chemokines. The axis length at each radius ranges from the minimum to maximum magnitude of log-abundance values across the analytes and the axis labels mark the log-abundance values at quartile intervals (0%, 25%, 50%, 75%, 100%). (**B**) Correlation analysis of chemokine expression with T cell infiltrate. Spearman correlation r values are plotted as a heatmap with the area of each circle corresponding to the strength of the correlation. Blue = positive correlation, Red = negative correlation. The correlation and statistical testing was performed using the rcorr function embedded in the Hmisc R package. TAC = tacrolimus.

**Figure 7 cancers-13-02131-f007:**
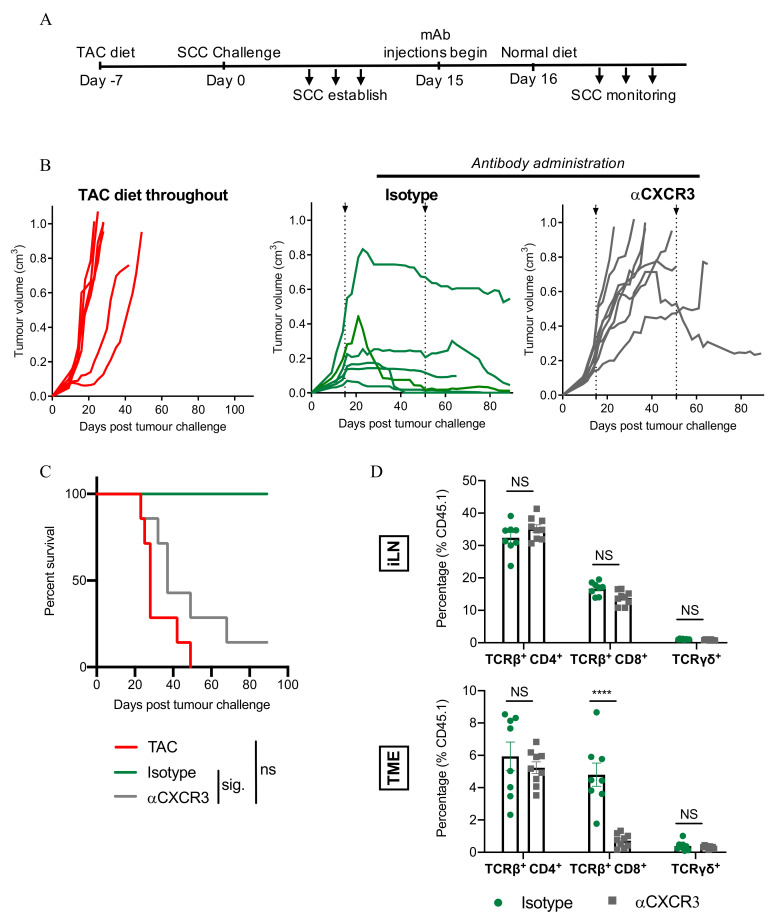
The CXCL10/CXCL11/CXCR3 axis plays a role in SCC regression. (**A**) Experimental schedule. (**B**) SCC growth pattern following treatment with the indicated antibodies. Combined data from 2 independent experiments, individual tumours shown, n = 6–8/group. (**C**) Survival comparisons of mice in B. Survival analyses were performed by Log-rank (Mantel-Cox) test. Results then displayed as significant (sig.) or not significant (ns) after bonferroni-corrected multiple comparison test. Bonferroni-corrected threshold = 0.016, K = 3. (**D**) Abundance of T cell subsets (%) in the inguinal lymph nodes (iLN) or in the tumour microenvironment (TME), as shown. Two-way ANOVA, bars represent mean ± SEM, data points represent individual mice. Data are combined from 2 independent experiments. *p* < 0.0001 (****), Dotted lines in (**B**) indicate the period of antibody administration.

**Figure 8 cancers-13-02131-f008:**
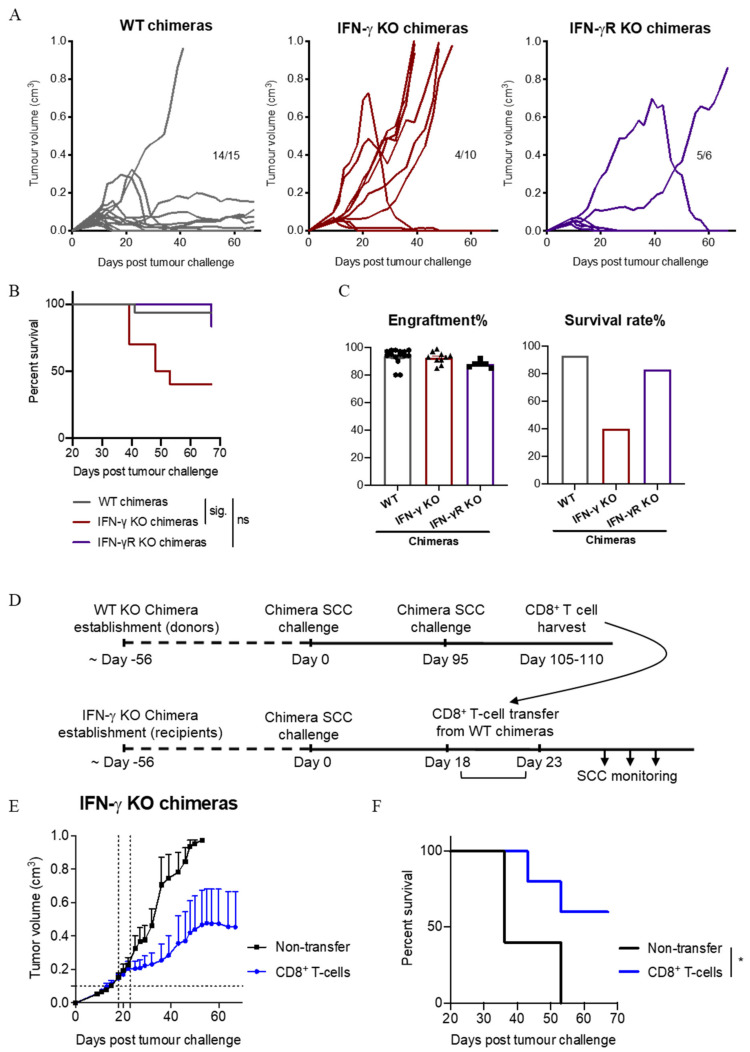
IFN-γ secretion by CD8^+^ T cells is important for SCC tumour control. (**A**) SCC establishment in individual chimeric mice. n = 15 WT, 10 IFN-γ^−/−^, and 6 IFN-γR^−/−^. Data shown are combined from 4 independent experiments. Numbers in panels represent the proportion of mice alive at the end of the experiment. (**B**) Comparison of mouse survival in A. Survival analyses was performed by Log-rank (Mantel-Cox) test. Results then displayed as significant (sig.) or not significant (ns) after bonferroni-corrected multiple comparison test. Bonferroni-corrected threshold = 0.016, K = 3. (**C**) Bone marrow engraftment efficiency and survival percentage in A. Data points represent individual mice. (**D**) Experimental schedule for CD8^+^ T cell transfer experiments into tumour-bearing IFN-γ KO chimeras. (**E**) Growth of established SCC tumours before and after CD8^+^ T cell transfer. T cells were transferred when individual tumours reached 0.1 cm^3^ (indicated by horizontal dotted line), which occurred between days 18 and 23 (represented by vertical dotted lines). Data shown combined from 2 independent experiments, mean ± SEM, n = 5/group. (**F**) Comparison of mouse survival in E. Survival analysis was performed by Log-rank (Mantel-Cox) test. *, *p* < 0.05.

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
