# Peer review of "IFN-γ Critically Enables the Intratumoural Infiltration of CXCR3+ CD8+ T Cells to Drive Squamous Cell Carcinoma Regression"

_cancers, 2021, doi:10.3390/cancers13092131_

Round 1
Reviewer 1 Report
IFN-g critically enabels the intratumoural infiltration of CXCR3+ CD8+ T cells to drive squamous carcinoma regression by Zeng at al.
Th authors investigated that the mechanisms of immune -mediated rejection of SCC using their established rejection model. After the removal of TAC, SCC grown in vivo was gradually regressed through the action of CD8+ T cells and IFN-g. The regression of SCC by IFN-g was mediated by the induction of chemokines for CXCR3, and blockade of CXCR3 signaling cancel the CD8+ T cell-dependent SCC regression. In addition, Deficiency of IFN-g, but not IFN-gR, in immune cells showed the progression of SSC in established mouse model. This study clearly showed the importance of IFN-g and their related chemokine for the activation of CD8+ T cells and the regression of SCC. However, I have an impression that the proposed mechanisms of immune responses concerning the regression of SSC (CD8+T cells, IFN-g, CXCR3 and their ligands) seem to be general findings, not new. Furthermore, some issues should be addressed as follow.
1) About the effect of IFN-γ produced by CD4+ T cells on SCC regression by CD8+ T cells. In Fig. 7A~C and Fig. 7D~E, the authors showed that hematopoietic cells-derived IFN-γ contributes to SCC control and that IFN-γ+/+ CD8+ T cells is important for SCC growth control. However, IFN-γ+/+ CD8+ T cells-transfer could not arrest tumor growth completely (about half level of non-transfer; Fig. 7E) and tumor mass size was larger than that of WT chimera transfer experiment (compared with Fig. 7A). In Fig. 1D, the effect of anti-TCRβ depletion was larger than that of anti-CD8β depletion. These data indicate the possibility that CD4+ T cells-derived IFN-γ affects SCC control by CD8+ T cells. Do CD4+ T cells support CD8+ T cells by producing IFN-γ? When CD4+ T cells from WT chimeras are additionally transferred to tumor-bearing IFN-γ-/- chimera in Fig. 7D (along with CD8+ T cells from WT chimeras), does tumor volume decrease further?
2) About the contribution of CXCR3-CXCL10 axis to CD8+ T cell migration. To reconfirm the contribution of CXCR3-CXCL10 axis to CD8+ T cell migration, authors had better use anti-CXCR3 Ab in WT chimera CD8+ T cells transfer experiment (Fig. 7D).
3) About CXCR3 expression on CD8+ T cells. How about CXCR3 expression on T cells of each chimera in Fig. 7A?
4) About IFN-gR KO chimera mouse. In figure 7A, different from IFN-g KO chimera mouse, IFN-gR KO chimera mouse displays the regression of SCC. In this case, are chemokine release also observed in tumor mass, the same as WT chimeras?
5) About chemokine release in the tumor microenvironment. In figure 5, the authors investigate the chemokine content in tumor mass. According to Materials and Methods section, the authors used harvested tumor mass, which contains immune cells, vascular endothelial cells, fibroblasts in addition to SCC. Which cells mainly produce chemokines in response to IFN-g in vivo? FACS analysis or immunohistochemistry are available for detection of chemokine producer cells.
6) About the contribution of CCL5 to SCC regression by CD8+ T cells. Fig. 5B indicates the correlation of CCL5 with CD8+ T cell infiltration. Although authors refer to the role of CCL5 as a further experiment in discussion, it is interesting to examine the expression of the receptor for CCL5 on tumor-migrating CD8+ T cells.
Author Response
1) About the effect of IFN-γ produced by CD4+ T cells on SCC regression by CD8+ T cells. In Fig. 7A~C and Fig. 7D~E, the authors showed that hematopoietic cells-derived IFN-γ contributes to SCC control and that IFN-γ+/+ CD8+ T cells is important for SCC growth control. However, IFN-γ+/+ CD8+ T cells-transfer could not arrest tumor growth completely (about half level of non-transfer; Fig. 7E) and tumor mass size was larger than that of WT chimera transfer experiment (compared with Fig. 7A). In Fig. 1D, the effect of anti-TCRβ depletion was larger than that of anti-CD8β depletion. These data indicate the possibility that CD4+ T cells-derived IFN-γ affects SCC control by CD8+ T cells. Do CD4+ T cells support CD8+ T cells by producing IFN-γ? When CD4+ T cells from WT chimeras are additionally transferred to tumor-bearing IFN-γ-/- chimera in Fig. 7D (along with CD8+ T cells from WT chimeras), does tumor volume decrease further?
We agree with the reviewer that CD4+ T cells may support CD8+ T cell activity within SCC tumours. We have included a new Figure 2 showing data from our FVB/SCC model that shows that a) there is an additional effect of depleting BOTH CD4+ T cells AND CD8+ T cells together, comparable to the effect of anti-TCRβ depletion (Fig 2A/B), and b) There is a small but significant increase in the percentage of CD4 T cells that express IFN-gamma within regressing SCC at both early rejection and later rejection time points (Fig 2C/D).
In coordination with this new figure, we have included the following new text into lines 276-314:
“CD4+ T cells contribute to the control of SCC growth
The disparity in SCC growth rate and consequently mouse survival when comparing TCRβ+ T cell depletion to CD8β+ T cell depletion (Figure 1) suggests a role for CD4+ T cells and possibly other immune subsets in combatting the rate of SCC growth. In a parallel set of experiments to those conducted in Figure 1, at the same time as TAC-diet withdrawal, mice with established SCC were injected with CD8β+ T cell-depleting antibody alone, or CD8β+ T cell-depleting antibody in combination with other subset-depleting antibodies (Figure 2A/2B). In the absence of CD8+ T cells, the majority of mice were unable to mediate SCC rejection, as expected. When comparing the effect of depleting CD4+ T cells, or TCRβ+ T cells, or TCRγδ+ T cells, or NK cells, or Gr-1+ myeloid cells in combination with CD8β+ T cells, three growth phenotypes emerged: 1) tumours grew faster (Figure 2B, left hand panels), or 2) tumours grew at a similar rate (Figure 2B, middle panels), or 3) tumours grew slower (Figure 2B, right hand panels). Combining CD8β+-depletion with either CD4+-depletion or TCRβ-depletion resulted in a faster rate of SCC growth and significantly re-duced survival when compared to CD8β+-depletion alone (Figure 2B, left hand panels). Notably, there were no significant differences in survival between ‘Anti-CD8β + Anti-CD4’ and ‘Anti-CD8β + Anti-TCRβ’ groups. Together, the data indicate a role for CD4+ T cells in controlling the rate of SCC growth. However, combining TCRγδ+ T cell or NK cell-depletion with CD8β+ T cell depletion did not show a significant effect on tumour growth or survival (Figure 2B, middle panels). In-terestingly, a slow-down in tumour growth (Figure 2B, top right panel) and improved survival (Figure 2B, bottom right panel) was apparent when ‘Anti-CD8β + Anti-Gr-1’ was compared to CD8β+-depletion alone, and two out of eight animals could reject established SCC tumours. Re-markably, the data indicate that the depletion of Gr-1+ myeloid cells allowed some mice to control tumour growth in the absence of CD8+ T cells, although ultimately, the majority of mice lacking CD8+ T cells were unable to reject established tumours. Taken together, the data suggest a positive role for CD4+ T cells and a negative role for Gr-1+ myeloid in the control of SCC growth.
IFN-γ is a well-known Th1 effector cytokine. To define whether CD4+ T cells, and CD8+ T cells, produce IFN-γ during SCC regression, mice were fed TAC-diet and challenged with SCC cells. Fourteen days later TAC diet was withdrawn, and tumours and draining lymph nodes were harvested after a further 7 days (day 21; when tumours were a similar size to control animals) or after a further 12 days (day 26; when tumours were visibly regressing, Figure 2C) and analysed for IFN-γ production by FACS. As shown in Figure 2D, following TAC removal there was a significant increase in the number of CD8+ T cells in the draining lymph nodes producing IFN-γ at day 21 but not by day 26. Conversely, in tumours, there was a variable but significant increase in IFN-γ-production by CD8+ T cells at day 21, and a strong and consistent production of IFN-γ by CD8+ T cells at day 26. CD4+ T cells, in contrast, did not produce detectible IFN-γ in the draining lymph nodes at either time point, but displayed a small but significant increase in the number of cells producing IFN-γ within the tumour mass at both time points. The results indicate that both CD8+ T cells and CD4+ T cells produce IFN-γ in the tumour environment following the removal of TAC.”
Unfortunately we did not look at the additional transfer of CD4+ T cells into chimeras at the same time as CD8+ T cells, but in light of the data above we speculate that tumour volume would decrease further.
2) About the contribution of CXCR3-CXCL10 axis to CD8+ T cell migration. To reconfirm the contribution of CXCR3-CXCL10 axis to CD8+ T cell migration, authors had better use anti-CXCR3 Ab in WT chimera CD8+ T cells transfer experiment (Fig. 7D).
We thank the reviewer for this suggestion, which is something that we had planned to conduct during the course of this study. However, due to the logistical complexity that CD8 transfer model requires and, more specifically, the low numbers of cells that we had available to transfer from WT chimeras, it was not possible to conduct this experiment. We hope that the data that we have supplied in our FVB/SCC model (now figure 7) will convince the reader of the importance of CXCR3 expression by CD8 T cells in bringing in CD8 T cells into the tumour microenvironment.
3) About CXCR3 expression on CD8+ T cells. How about CXCR3 expression on T cells of each chimera in Fig. 7A?
We think that the reviewer is asking whether IFN-gamma neutralisation has an effect on CXCR3 expression on T cells. We did not perform this analysis for the chimera work because we had previously established that IFN-gamma neutralisation has no effect on T cell CXCR3 expression in our FVB/SCC model. We have now included this data as a new Supplementary Figure 4 and made the following addition to the text in line 419 “but had no impact on CXCR3 expression by T cell subsets (Figure S4A/B).”
4) About IFN-gR KO chimera mouse. In figure 7A, different from IFN-g KO chimera mouse, IFN-gR KO chimera mouse displays the regression of SCC. In this case, are chemokine release also observed in tumor mass, the same as WT chimeras?
The reviewer is correct that one mouse displayed regression of SCC in the IFN-gR KO chimera model, however in one mouse the SCC grew and did not regress (SCC were unable to establish in the remaining 4 mice). Importantly, both mice displayed mixed intervals of tumour growth and tumour regression. Similar growth patterns are also present in the WT chimera model. While it would be technically possible to look at chemokines in the tumours of these mice if one were to substantially increase the numbers of animals used, we could not say with any confidence whether the tumours that we harvest would have ultimately progressed or regressed, thus confusing our interpretation of the data. Therefore, given the consistency of the progression/regression outcomes provided by the FVB/SCC model, we feel that the interpretation of chemokine release in this model (as supplied in Figure 6) is more informative and carries a lower risk of misinterpretation.
5) About chemokine release in the tumor microenvironment. In figure 5, the authors investigate the chemokine content in tumor mass. According to Materials and Methods section, the authors used harvested tumor mass, which contains immune cells, vascular endothelial cells, fibroblasts in addition to SCC. Which cells mainly produce chemokines in response to IFN-g in vivo? FACS analysis or immunohistochemistry are available for detection of chemokine producer cells.
The reviewer is of course correct and indeed there are many many more subtypes of cells present within the tumour mass over and above those mentioned above. A FACS or immunohistochemistry analysis would rely on a strong knowledge of all the subtypes present and furthermore, an array of markers that could be used with confidence to delineate each subtype. While clearly an important area of research, to do this analysis well deserves a targeted and complete study in its own right in order to perform the necessary ground work and optimisation, and is therefore beyond the scope of the current study.
6) About the contribution of CCL5 to SCC regression by CD8+ T cells. Fig. 5B indicates the correlation of CCL5 with CD8+ T cell infiltration. Although authors refer to the role of CCL5 as a further experiment in discussion, it is interesting to examine the expression of the receptor for CCL5 on tumor-migrating CD8+ T cells.
This is indeed an interesting experiment, however there are 4 known receptors for CCL5 (CCR1, CCR3, CCR5, GPR75), and indeed there may even be more. The interpretation of the data may therefore not be straightforward and would certainly require some specialist knowledge of the receptors themselves and whether or not they display a hierarchy of engagement or even a time-line of expression. While interesting, we feel that the analysis of CCL5 receptors is best performed as part of our next study.
Reviewer 2 Report
The manuscript is clear and easy to read. The experiments show clearly what is stated in the abstract.
Major comments:
_ In Fig. 4E, the anti INF treatment leads to less CXCR3+ CD4+ T cells in the TME. But in Fig. 6D, there is no significant difference between the number of CD4+ T cells after anti-CXCR3 administration. The authors need to comment in the text on this apparent discrepancy.
_ The story is spotless until we get to Fig. 7, where things become a bit more complicated. Figure 7A suggests that INF is not produced by the tumour cells but instead by hematopoietic cells. It would then suggest a scenario where some unidentified hematopoietic cells (DCs, NK T cells, macrophages?) are recruited to the TME, where they produce INF. This production of IFN would then promote secretion of CXCL10 (Fig. 3E) by SCC cells, which in turn would recruit CD8+ CTLs and lead to tumour regression.
The problem is that we have a chicken and egg problem if these hematopoietic cells are T cells (fig. 7E and F): T cells are recruited to the TME to secrete INF. But to be recruited to the TME, INF must be secreted so that SCC cells express CXCL10 and recuit T cells. There is a loop where T cells must already by at the TME so that T cells are recruited to the TME. The authors need to provide some sort of explanation to this conundrum and make hypothesis based on current literature regarding the nature of the INF-secreting cells at the TME in their model.
Minor comments:
_ In fig. 4 it would be good to explain why the cell number are expressed as a percentage of CD45.1
_ Line 400, I think it’s Fig. 6D instead of 5D
_ Some sort of schematic of the experimental design of transfer approach in Fig. 7 would ease the understanding of these experiments
Author Response
Major comments:
_ In Fig. 4E, the anti INF treatment leads to less CXCR3+ CD4+ T cells in the TME. But in Fig. 6D, there is no significant difference between the number of CD4+ T cells after anti-CXCR3 administration. The authors need to comment in the text on this apparent discrepancy.
We have now added the following text into the discussion (lines 601-611):
“The observation that an IFN-γ neutralising antibody can block CXCR3+CD4+ T cell in-filtration into the tumour mass (Figure 5E), but an antibody that blocks the binding of CXCL10 and CXCL11 to CXCR3 cannot (Figure 7E), suggests that while CXCL10 is im-portant for the infiltration of CD8+ T cells into SCC, different chemokines may influence the migration of CD4+ T cells into SCC. In addition to CXCL10, the injection of anti-IFN-γ also significantly impaired the production of CXCL9 and CCL5 in the tumour microenvi-ronment compared to isotype antibody-treated tumours (Figure S3). Our data also indicate a strong positive correlation between CCL5 and CD4+ T cell abundance in tumours (Figure 6B and Table S1). Therefore, IFN-γ also appears to play an important role in the infiltra-tion of CD4+ T cells into SCC, possibly through the induction of CCL5.”
_ The story is spotless until we get to Fig. 7, where things become a bit more complicated. Figure 7A suggests that INF is not produced by the tumour cells but instead by hematopoietic cells. It would then suggest a scenario where some unidentified hematopoietic cells (DCs, NK T cells, macrophages?) are recruited to the TME, where they produce INF. This production of IFN would then promote secretion of CXCL10 (Fig. 3E) by SCC cells, which in turn would recruit CD8+ CTLs and lead to tumour regression.
The problem is that we have a chicken and egg problem if these hematopoietic cells are T cells (fig. 7E and F): T cells are recruited to the TME to secrete INF. But to be recruited to the TME, INF must be secreted so that SCC cells express CXCL10 and recuit T cells. There is a loop where T cells must already by at the TME so that T cells are recruited to the TME. The authors need to provide some sort of explanation to this conundrum and make hypothesis based on current literature regarding the nature of the INF-secreting cells at the TME in their model.
We have now added the following text into the discussion (lines 582-597):
“Using our chimeric models, we have established the importance of IFN-γ produced by hematopoietic cells in the control of SCC, and furthermore, that IFN-γ produced by CD8+ T cells plays a role. In other experiments we have also established that CXCL10 production is key for CD8 T cell-mediated SCC rejection. While it is feasible to argue that IFN-γ produced by CD8+ T cells within the tumour acts in a positive-feedback loop to in-duce further CXCL10 expression from SCC cells, leading to further CD8+ T cell infiltration, this loop does not explain the initiating, hematopoietic, source of IFN-γ within the tumour microenvironment that led to CXCL10 induction in the first place. Many innate immune cells are known to release IFN-γ, including NK cells, NKT cells, γδ T cells, dendritic cells and macrophages [25], and co-culture experiments in the laboratory have confirmed that SCC cells can directly stimulate IFN-γ release by NK cells [41]. Therefore, it is possible that innate immune cells, such as NK cells, provide an early source for IFN-γ. Although it is interesting to note that the depletion of NK cells (or γδ T cells or Gr-1+ mono-cytes/macrophages/dendritic cells) cells did not prevent tumour regression in our model (Figure 1D), it is possible that IFN-γ had already been released by these cells into the tu-mour microenvironment prior to their depletion on day 9.”
Minor comments:
_ In fig. 4 it would be good to explain why the cell number are expressed as a percentage of CD45.1
To explain this point we have now made the following additions (underlined) to the text for this figure legend (lines 406-408):“(B-E) Data are derived from 3 independent experiments, n=6-10/group, and are normalised to CD45.1 expression to mitigate the impact caused by large differences in total cell counts when comparing mice with regressing and non-regressing tumours. Statistical analysis was performed by unpaired Student’s t-test.
_ Line 400, I think it’s Fig. 6D instead of 5D
Yes, you are quite correct – thank-you for pointing out this typo. We have now corrected it to 7D in line with the updated Figure numbering following the addition of additional figure earlier in the manuscript.
_ Some sort of schematic of the experimental design of transfer approach in Fig. 7 would ease the understanding of these experiments
We have now expanded our experimental design schematic for the transfer approach in Figure 8D to improve the understanding of the complexity of this experiment.
Reviewer 3 Report
I have reviewed the manuscript entitled” IFN-γ Critically Enables the Intratumoural Infiltration of CXCR3+ CD8+ T Cells to Drive Squamous Cell Carcinoma Regression” by Zeng Z et al. In this paper, the authors have developed a cell line from an SCC tumor that appeared following the chronic UV-treatment of HPV38E6E7 engineered mice to express the Human Papilloma Virus 38 E6 and E7 oncogenes under the control of the Keratin 14 promoter. The aim of this study was to identify the key immune subsets that mediated rejection of SCC, and to elucidate the mechanistic role of the Interferon-gamma in this process. The data regarding the SCC regressor tumor model in which SCC grown in mice immunosuppressed with tacrolimus are subsequently destroyed by the immune system following the removal of immunosuppression, permitting an examination of the role played by IFN-γ in the rejection of SCC. This report provides a way herein to find out the certain IFN-γ mediated CXCR3/CXCL10/11 chemokine axis in CD8+ T cell-mediated SCC tumor regression is novel and interest. However, several major comments can however be made:
- If any NGS analysis for that HPV38 SCC regressor cell line to that related SCC progressed cell line? What’s the mutagenesis occurred with that HPV38 SCC which might be responded to IFN-γ mediated immunity?
- Neutralized Abs which specific reacted to CD4-, CD8β-, TCRβ-, TCRγδ-, Gr-1, Asialo-GM1 was individually administrated into TAC diet infused HPV38 SCC-bearing mice to explore the immune mediators of SCC tumor regression in Fgure1. The data showed that CD8+ T cells, but not the CD4+ T cells, might be the majority player involved in the progression and regression of SCC development. Thus, the regression of SCC can be also abrogated by IFN-γ neutralization (Figure 2), of which might be mediated by certain chemokines (CXCL10) secretion of SCC cells (Figure 3) and the engagement of their appropriated receptors on CD4/CD8 T cells (Figure 4). Therefore, IFN-γ neutralization not only reduced the number of CXCR3+CD8+ but also the frequency of CXCR3+CD4+ T cell subsets that infiltrate into SCC tumors. If the IFN-γ critically enables the intra-tumoral infiltration of CD8+CXCR3+ and CD4+CXCR3+ T cell subsets into tumor mass, what’s the role of CD4+CXCR3+ T cells under IFN-γ to response to tumor? The tumor volume in that anti-TCRβ treated group was higher than that anti-CD8β-administrated mice, it might still indicate that there was certain role which may contribute by CD4 T cells. How about the combination of anti-CD4 and anti-CD8 Abs to SCC regression?
- The role of IFN-γ to the production of CXCL10 in SCC and recruitment of CD4 and/or CD8 T cells to SCC cells through regulation of CXCR3 expression onto various T subsets should be validated in vitro. The anti-CXCR3 Abs affected the SCC regression significantly reduced the CD8+ T cells but not CD4+ T cells in vivo. What’s the better interpretation to compare the reduction of CXCR3+CD4+ T cells in the IFN-γ neutralization TME of SCC? The Anti-IFN-γ treatment significantly impaired the production of CXCL9, CXCL10 and CCL5 when compared to isotype-treated tumor (Figure S3). If this data indicated that the chemokine usage was distinguished to various CD4 and CD8 T subsets?
- The significance of IFN-γ producing CD8+ for SCC tumor control was evaluated in Figure 7, how about the IFN-γ producing CD4+ alone and the combination of IFN-γ producing CD8+ /CD4+ T cells in IFN-g/R KO kimeras?
Author Response
- If any NGS analysis for that HPV38 SCC regressor cell line to that related SCC progressed cell line? What’s the mutagenesis occurred with that HPV38 SCC which might be responded to IFN-γ mediated immunity?
Unfortunately, no NGS analysis exists for the SCC regressor cell line, although as the cell line is derived from a UV-induced tumour and the outgrowth of transferred cells can be controlled in healthy mice we would envisage that there would be a large cohort of immunogenic neoantigens that are driving the T cell response.
- Neutralized Abs which specific reacted to CD4-, CD8β-, TCRβ-, TCRγδ-, Gr-1, Asialo-GM1 was individually administrated into TAC diet infused HPV38 SCC-bearing mice to explore the immune mediators of SCC tumor regression in Fgure1. The data showed that CD8+ T cells, but not the CD4+ T cells, might be the majority player involved in the progression and regression of SCC development. Thus, the regression of SCC can be also abrogated by IFN-γ neutralization (Figure 2), of which might be mediated by certain chemokines (CXCL10) secretion of SCC cells (Figure 3) and the engagement of their appropriated receptors on CD4/CD8 T cells (Figure 4). Therefore, IFN-γ neutralization not only reduced the number of CXCR3+CD8+ but also the frequency of CXCR3+CD4+ T cell subsets that infiltrate into SCC tumors. If the IFN-γ critically enables the intra-tumoral infiltration of CD8+CXCR3+ and CD4+CXCR3+ T cell subsets into tumor mass, what’s the role of CD4+CXCR3+ T cells under IFN-γ to response to tumor? The tumor volume in that anti-TCRβ treated group was higher than that anti-CD8β-administrated mice, it might still indicate that there was certain role which may contribute by CD4 T cells. How about the combination of anti-CD4 and anti-CD8 Abs to SCC regression?
The reviewer has raised some important points that we have addressed through the addition of new data. In a new Figure 2, we now show the effects of the combination of anti-CD4 and anti-CD8 as suggested by the reviewer. In the data it can be seen that the rate of tumour growth increases when the deletion of both CD4 and CD8 is compared to the deletion of CD8 alone. Therefore, it appears likely that CD4 T cells play a role in the control of SCC growth, as we outline in the new results text that corresponds to the figure. Furthermore, also in the new data (Figure 2D) we show that CD4 T cells within regressing tumours show a small but significant increase in the production of IFN-γ. Therefore, it is likely that CD4 T cells contribute to the general anti-tumour immune response within tumours, at least in part, as an additional source of IFN-γ
- The role of IFN-γ to the production of CXCL10 in SCC and recruitment of CD4 and/or CD8 T cells to SCC cells through regulation of CXCR3 expression onto various T subsets should be validated in vitro. The anti-CXCR3 Abs affected the SCC regression significantly reduced the CD8+ T cells but not CD4+ T cells in vivo. What’s the better interpretation to compare the reduction of CXCR3+CD4+ T cells in the IFN-γ neutralization TME of SCC? The Anti-IFN-γ treatment significantly impaired the production of CXCL9, CXCL10 and CCL5 when compared to isotype-treated tumor (Figure S3). If this data indicated that the chemokine usage was distinguished to various CD4 and CD8 T subsets?
We think the reviewer is asking whether IFN-γ impacts the migration of CD4 T cells through chemokines other than those that act as ligands for CXCR3. This is entirely possible since our data in Figure 6A/S3 show that IFN-γ-neutralisation affects the abundance of CCL5 and CXCL9 within the tumour mass, and our data in Figure 6B/Table S1 show a positive correlation between CCL5 and CD4 T cell abundance. We cannot rule out a role for CXCL9 here in this observation because the antibody that we have used to block CXCR3 does not prevent CXCL9 from binding to CXCR3, as we mention in the paper. Therefore, it may well be the case that CD8 T cells rely heavily on CXCL10 to migrate into the tumour, whereas CD4 T cells rely more on CCL5. We have now added the following text to the discussion to address this point (lines 601-611):
“The observation that an IFN-γ neutralising antibody can block CXCR3+CD4+ T cell in-filtration into the tumour mass (Figure 5E), but an antibody that blocks the binding of CXCL10 and CXCL11 to CXCR3 cannot (Figure 7E), suggests that while CXCL10 is im-portant for the infiltration of CD8+ T cells into SCC, different chemokines may influence the migration of CD4+ T cells into SCC. In addition to CXCL10, the injection of anti-IFN-γ also significantly impaired the production of CXCL9 and CCL5 in the tumour microenvi-ronment compared to isotype antibody-treated tumours (Figure S3). Our data also indicate a strong positive correlation between CCL5 and CD4+ T cell abundance in tumours (Figure 6B and Table S1). Therefore, IFN-γ also appears to play an important role in the infiltra-tion of CD4+ T cells into SCC, possibly through the induction of CCL5.”
- The significance of IFN-γ producing CD8+ for SCC tumor control was evaluated in Figure 7, how about the IFN-γ producing CD4+ alone and the combination of IFN-γ producing CD8+ /CD4+ T cells in IFN-g/R KO kimeras?
The suggestion to transfer IFN-γ producing CD4 T cells into IFN-γ KO chimeras is an intriguing one, but as can be seen in new Figure 2D, no increase could be found in CD4 T cell IFN-γ production within the lymph nodes of mice with control or regressing tumours, at either an early rejection time point or a later rejection time point during regression. Since we rely upon the secondary lymphoid organs as a source of T cells for transfer, and the data suggest that IFN-γ producing CD4 are not present in the lymph nodes, the transfer of CD4 T cells that we could reasonably expect to produce IFN-γ in response to SCC is not technically feasible. It would be technically feasible to transfer IFN-γ producing CD8+ T cells into IFN-g/R KO chimeras however, as we have done into IFN-γ KO chimeras, but IFN-g/R KO chimeras already have their own complement of IFN-γ producing CD8+ T cells (it is only the IFN-g/R CD8 T cells that they lack). Furthermore, the interpretation of the outcomes that this transfer would have would be very difficult to make given that 4 out of 6 IFN-g/R KO chimeras are able to prevent tumour establishment without such a transfer.
Round 2
Reviewer 1 Report
This reviewer was satisfied with author's revision.
Thank you for your sincere responses.
Reviewer 3 Report
The message of the manuscript is now straight forward